# Optimization of a Green Extraction of Polyphenols from Sweet Cherry (*Prunus avium* L.) Pulp

Maria Lisa Clodoveo [1,†] , Pasquale Crupi [1,*,†] and Filomena Corbo [2]

1 Interdisciplinary Department of Medicine, University Aldo Moro Bari, 70125 Bari, Italy
2 Department of Pharmacy-Drug Sciences, University Aldo Moro Bari, 70125 Bari, Italy
* Correspondence: pasquale.crupi@uniba.it; Tel.: +39-3471252849
† These authors contributed equally to this work.

**Abstract:** This work focused on the optimization of the ultrasound (US) extraction of polyphenols from sweet cherry pulp by monitoring cyanidin-3O-rutinoside, quercetin-3O-rutinoside, and *trans*-3-O-coumaroylquinic acid, representing the main anthocyanin, flavonol, and hydroxycinnamate, respectively, identified in the extracts through chromatographic analyses (HPLC-DAD), as output variables. The optimization was performed following a two-level central composite design and the influence of the selected independent variables (i.e., extraction time and solid to solvent ratio) was checked through the response surface methodology. The maximum recovery of the phenolic compounds was obtained at 3 min and 0.25 g/mL in water/ethanol (1:1, *v/v*) at a set temperature (25 °C), sonication power (100 W), and sonication frequency (37 kHz). Subsequent validation experiments proved the effectiveness and reliability of the gathered mathematical models in defining the best ultrasound-assisted extraction conditions.

**Keywords:** cv. Ferrovia; DoE; energy saving; green solvent; organic raw material; UAE





## 1. Introduction

*Prunus avium* L., commonly known as sweet cherries, are very appreciated red fruits, which in numerous areas of Europe, West Asia, and South America appear on the market early in the season [1]. Their production is mainly exploited in the fresh fruit market, even though particular climatic conditions (such as the strong incidence of rainfall during the harvest period) can negatively affect their integrity and quality, making them unsaleable as fresh fruit but suitable to be processed into other products such as juice, jam, marmalade, and toppings [2,3].

Due to their acknowledged significant content of polyphenols (i.e., flavonoids and hydroxycinnamates) possessing a lot of biological activities [4–6], the production leftovers of sweet cherries (ranging from 15–30% of the total harvesting) can be valorized through the extraction of these antioxidant compounds for their incorporation into functional foods, pharmaceutical drugs, cosmetic, dietary supplements, and nutraceuticals, while also contributing to the circular economy [2,7].

Currently, a hot research topic consists in applying suitable techniques for the extraction of bioactive compounds from fruit matrices, which are quantitative, non-destructive, and time-saving [3,8–10]. In this sense, green extraction, based on the use of alternative solvents (i.e., water, ethanol, or their mixture) and renewable natural products and setting more rapid extraction processes with reduced unit operations and energy consumption, perfectly fits the aforementioned purpose [11,12].

Energy consumption can be reduced, for instance, by optimizing the extraction processes operating parameters (i.e., extraction time and solvent-solid ratio), particularly through innovative technologies (i.e., US). In many cases, these innovative technologies appear energy-efficient since they enable to reach maximal yields in reduced extraction

time [13,14]. Moreover, the remarkable growth of the circular economy will rely on new technologies moving toward sustainable processes [15]. Ultrasound-assisted extraction (UAE) is an environmentally friendly extraction method that offers better reproducibility, simpler and less expensive equipment, and lower solvent and energy consumption compared to the conventional extraction technique [16,17].

In order to find the best conditions to maximizing the recovery of bioactive compounds, the use of optimization models is essential [18]. Response surface methodology (RSM), which allows simultaneous optimization of the individual factors along with their possible interactions by providing a polynomial equation that fits the experimental data, is usually preferred to the more time-consuming one-factor-a-time approach [19].

To the best of our knowledge, only two recent reports refer to optimizing US extraction methods for polyphenols from sweet cherries: the former specifically dealing with anthocyanins [18] while the latter uses a petroleum-derived solvent [20], and both employing conventional methods with prolonged extraction times. Therefore, hypothesizing to conduct effective and efficient extraction with reduced time, this study aimed to optimize UAE parameters (i.e., extraction time and solid to solvent ratio, using water/ethanol 1:1, *v/v*), influencing the extraction of the main polyphenols identified by HPLC-DAD analysis, through a central composite design (CCD) coupled to the RSM tool. Since, another principle involved in green extraction regards the adoption of raw materials from food crops grown under controlled cultivation conditions (water, pesticides, and fertilizers), the sweet cherry (cv. Ferrovia) selected for this work was from organic farming.

## 2. Materials and Methods

### 2.1. Plant Materials

The trial was carried out in 2021 on mature sweet cherries (*Prunus avium* L.) of the Ferrovia variety, collected from seven-year-old trees located in Turi (longitude 40.56° E, latitude 17.12° N, and altitude 280 m) and cultivated under organic farming. The trees were trained to a central leader system and planted at a spacing of 4 m × 4 m; starting from blossoming and until the harvest, they were irrigated through a localized system (drip irrigation) with a water supply of 600 m³/ha. Ferrovia is a cultivar with medium-low vigor, rising-spreading habit, medium productivity, uniform distribution of the fruits on the plant. It has large-sized fruits (average weight 8.2 g) with long peduncles. The peel is bright red, the pulp is very firm and pink in color (albeit darker in color near the stone).

Samples were harvested (second decade of June) once reached the best organoleptic quality parameters, in particular total soluble solids (TSS), measured as °Brix, and titratable acidity (TA), expressed as a percentage of malic acid (TSS = 15.1 ± 1.1 °Brix; TA = 0.84 ± 0.11%).

A total of 1 kg of cherries was taken on the same day, from four different branches of an individual tree and mixed, and then, they were frozen in liquid nitrogen and vacuum packed in plastic bags and stored at −80 °C for further analysis.

### 2.2. Chemicals

Formic acid, ethanol, and HPLC grade water and acetonitrile were supplied from Merk Life Science S.r.l. (Milano, Italy). Chlorogenic acid was purchased from Phytolab (Aprilia, Italy), cyanidin-3-O-rutinoside chloride, and quercetin-3-O-rutinoside were purchased from Extrasynthese (Genay, France) and used as HPLC reference standards.

### 2.3. Ultrasound-Assisted Extraction of Polyphenols from Cherries

Roughly 30 g of destoned sweet cherry berries were freeze-dried (CHRIST Alpha 1–4 LD plus, Osterode am Harz, Germany) at −42 °C under a pressure of 0.10 mbar for 72 h and ground using an IKA A11 basic homogenizer (IKA®-Werke GmbH & Co., Staufen, Germany). 250 mg of powder samples, properly sieved at 0.5 mm (Endecotts Ltd., London, UK) to obtain uniformly sized particles and carefully weighed (EU-C1200, Gibertini s.r.l., Novate Milanese, Milano, Italy) into 2 mL Eppendorf tubes in presence of water/ethanol

(1:1, *v/v*), were used in each experiment according to the extraction conditions in the experimental design. An ultrasonic water bath (Elmasonic P 30H, Elma Schmidbauer GmbH, Singen, Germany) working in continuous mode at fixed temperature (25 ± 2 °C) power (100 W), and frequency (37 kHz), was employed for UAE of polyphenols.

Afterward, the extracts were centrifuged at 4000× *g* for 15 min at 5 °C (EPPENDORF 5810R, Hamburg, Germany), filtered through a 0.45 μm syringe cellulose filter, and analyzed by HPLC-DAD.

### 2.4. HPLC-DAD Analysis

Polyphenols analyses were carried out through an HPLC 1260 (Agilent Technologies, Palo Alto, CA, USA), composed of a degasser, quaternary pump (model G1311C), column compartment (model G1316A), and diode array detector (model G1315D). The extracts (3 μL) were injected onto a reversed stationary phase column, Luna C18 (150 × 2 mm i.d., particle size 3 μm, Phenomenex, Torrance, CA, USA) using a model G1329B autosampler (Agilent Technologies, Palo Alto, CA, USA). The main column was protected by Gemini C18 (Phenomenex, Torrance, CA, USA) 5 μm (4 × 2 mm i.d.) pre-column and maintained at 40 °C. The following binary gradient, column re-equilibration, and flow rate were the same as those reported in our previous research [4]. Diode array detection was between 190 and 650 nm, and absorbance was recorded at 520, 360, and 320 nm.

Tentative compound identification was achieved by comparing retention times (RT) and absorption spectra profile and maxima ($\lambda_{max}$) with those reported in our previous research [4,5]. Quantification of polyphenols was made by using the calibration curves in the concentration range 10–200 μg/mL of cyanidin-3-O-rutinoside ($R^2$ = 0.9964; LOD = 4.46 μg/mL; LOQ = 14.9 μg/mL), chlorogenic acid ($R^2$ = 0.9983; LOD = 3.04 μg/mL; LOQ = 10.1 μg/mL), and quercetin-3-O-rutinoside ($R^2$ = 0.9994; LOD = 1.9 μg/mL; LOQ = 6.3 μg/mL). The detection limit (LOD) and quantification limit (LOQ) were calculated from the calibration curves (3 and 10 folds, respectively, the ratio between intercept error and slope).

### 2.5. Experimental Design and Statistical Analyses

A two-factor standard CCD was performed to optimize the effect of extraction time ($X_1$) and solid-solvent ratio ($X_2$) on the extraction of polyphenols from sweet cherries pulp. Twelve randomized experiments were conducted, consisting of four cube points, four star points, and four replicates at the center values to evaluate the pure error sum of squares and lack of fit test (Figure 1).

CCD was designed and analyzed by RSM using STATISTICA 12.0 software package (StatSoft Inc., Tulxa, OK, USA). Three second-order polynomial equations (quadratic model) were developed to fit the experimental raw data of the flavonoids and hydroxycinnamate:

$$Y_i = B_0 + \Sigma B_i X_i + \Sigma B_{ii} X_i^2 + \Sigma B_{ij} X_i X_j$$

where $Y_i$ is the response function of each output variable; $B_0$ is a constant coefficient; $B_i$ are the regression coefficients of the linear, quadratic, and interactive terms, and $X_i$, $X_j$ represent the independent variables ($X_1$ and $X_2$). After analysis of variance (ANOVA), the regression coefficients of the individual linear, quadratic, and interaction terms were determined and the fit of the mathematical models was performed by evaluating the $R^2$ and $R^2_{adj}$. Subsequently, the profiler tool was used to produce, through a general desirability function optimization procedure, the prediction profile graph that shows the relationship between predicted responses on the three polyphenols and the desirability of responses. Finally, further extraction trials, obtained in triplicates under the optimized UAE, were carried out for the model validation and the experimental data were compared to the predicted concentrations by a *t*-test for single means.

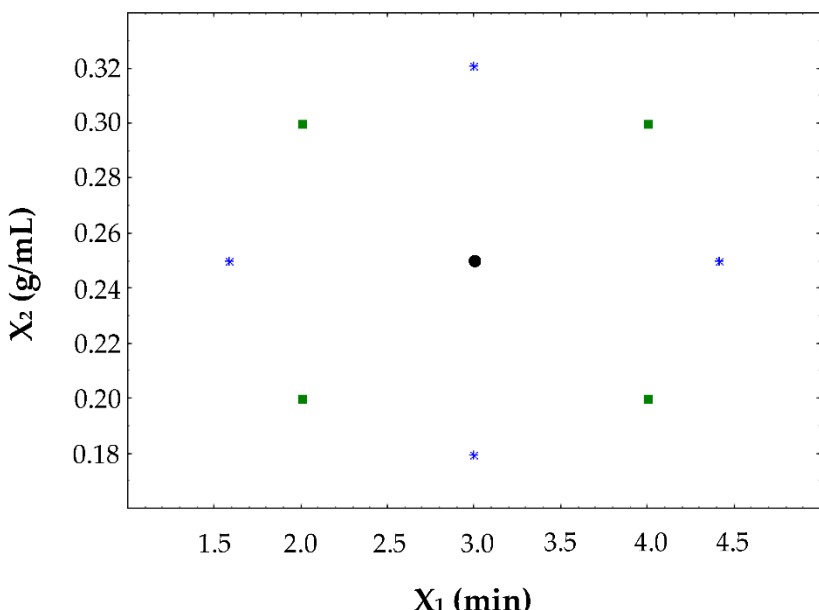

**Figure 1.** CCD features: four cube points (green squares), four star points (blue crosses), and four center points (black circles). $X_1$: extraction time. $X_2$: solvent-solid ratio.

## 3. Results and Discussion

### 3.1. HPLC-DAD Identification of Polyphenols in Cherry Pulp Extract

Figure 2 reports the chromatograms registered at 320, 360, and 520 nm of a cherry pulp extract in water/ethanol (1:1, *v/v*). Considering the similarity of chromatographic conditions, in particular, the stationary phase and gradient program used, to those of our previous relevant works [4,5], the polyphenols identification was accomplished by matching the retention times, elution order (confirmed by means of the reference standards), and UV/Vis spectra. Therefore, 13 peaks were assigned to flavonoids (including anthocyanins and flavonols), and 17 were attributed to hydroxycinnamates derivatives (Table S1—Supporting Information).

According to the amounts already found in cv. Ferrovia cherries [4,5,21], *trans*-3-O-coumaroylquinic acid, quercetin-3-O-rutinoside, and cyanidin-3-O-rutinoside were the main compounds (as hydroxycinnamate, flavonol, and anthocyanin, respectively) revealed in our extracts (Figure 2, Table S1). Thus, they were selected as output variables to define the optimal parameters for UAE.

### 3.2. Optimization of UAE Conditions by CCD-RSM

According to their structure or their interactions with other fruit components, various extraction conditions (i.e., time, solvent type, and temperature) were employed in the literature [3,20]. It is acknowledged that prolonged extraction time, aqueous solvents, and especially high temperatures (>50 °C) might provoke chemical oxidation of polyphenols due to OH· species whose formation would be facilitated by the US [13]. In this work, to adopt a procedure as closely as possible to green extraction purposes [12], a UAE procedure was followed by maintaining the extraction temperature at 25 °C and using a mix of water/ethanol (1:1, *v/v*) as solvent type, while, on the basis of previous screening experiments (data not shown), the extraction time and solid-liquid ratio were set in the range 2–4 min and 0.2–0.3 g/mL, respectively, to allow a reduced energy and solvent consumption. It is also worth pointing out that the choice of organic sweet cherries (cv. Ferrovia) was strictly associated with the need for natural extracts, which should be biodegradable and without contaminants [11]. A standard CCD (with $\alpha_O$ = 1.4142 and $\alpha_R$ = 1.0781 for orthogonality and rotatability, respectively) was developed to optimize $X_1$ and $X_2$ factors affecting US extraction of phenolic compounds from the sweet cherry pulp.

The concentrations of the 3 polyphenols (expressed in µg/mL) and the natural values of the factors for the 12 experiments, which were performed randomly to obtain an accurate estimate of the experimental error, are shown in Table 1.

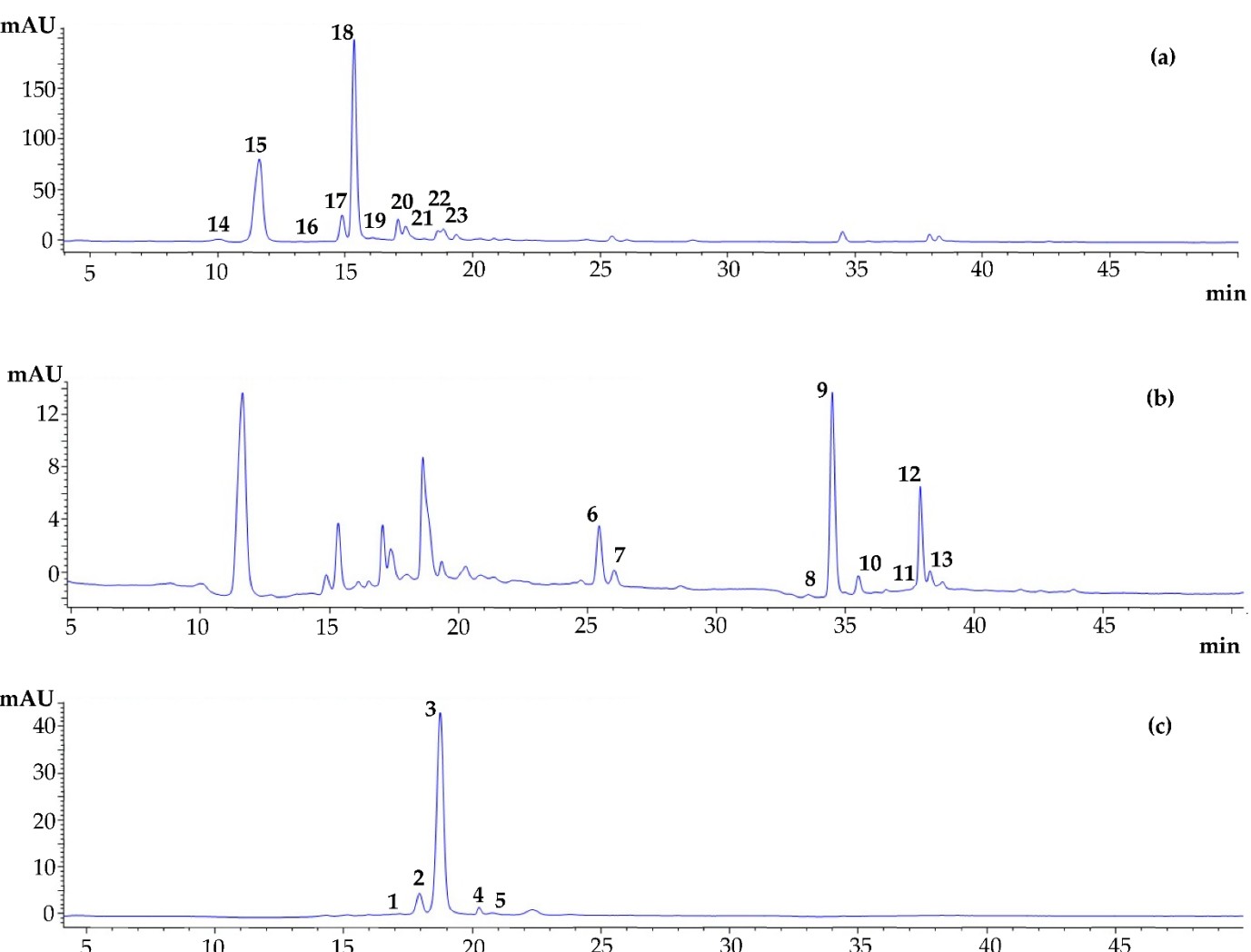

**Figure 2.** DAD chromatograms at (**a**) 320 nm, (**b**) 360 nm, and (**c**) 520 nm of cherry pulp (cv. Ferrovia) extract. Peaks number corresponds to the compounds listed in Table S1.

Table 2 lists the second-order predictive polynomial equations obtained by applying the quadratic regression models to the experimental values to portray the empirical relationship between polyphenol concentrations and operating conditions. The non-significance ($p > 0.05$) of the model's lack of fit test, which was performed by repeating the observations four times at the central point, proved the reliability of the obtained polynomials. The coefficients of determination ($R^2$) were generally > 0.9 (except in the case of quercetin-3-O-rutinoside), indicating that only < 10% of the total variation was not explained by the models and that there was an overall good degree of correlation between the observed and predicted values. The adjusted coefficients of determination ($R^2_{adj}$) were close to $R^2$, confirming good statistical models (Table 2).

**Table 1.** UAE optimization by two-level central composite design (CCD).

| Experiments | $X_1$ (min) | $X_2$ (g/mL) | Cyanidin-3-O-Rutinoside (µg/mL) | Quercetin-3-O-Rutinoside (µg/mL) | *trans*-3-O-Coumaroylquinic Acid (µg/mL) |
|---|---|---|---|---|---|
| 6 [a] | 4.41 | 0.25 | 61.8 | 10.9 | 79.3 |
| 3 [b] | 4.00 | 0.20 | 49.2 | 10.2 | 57.2 |
| 11 (C) [c] | 3.00 | 0.25 | 61.9 | 10.7 | 79.8 |
| 10 (C) [c] | 3.00 | 0.25 | 64.2 | 12.7 | 85.4 |
| 4 [b] | 4.00 | 0.30 | 49.7 | 11.6 | 73.3 |
| 12 (C) [c] | 3.00 | 0.25 | 56.1 | 10.6 | 80.7 |
| 5 [a] | 1.59 | 0.25 | 64.6 | 11.3 | 75.7 |
| 1 [b] | 2.00 | 0.20 | 37.5 | 8.2 | 54.6 |
| 9 (C) [c] | 3.00 | 0.25 | 60.6 | 11.5 | 83.6 |
| 2 [b] | 2.00 | 0.30 | 53.0 | 12.5 | 79.8 |
| 7 [a] | 3.00 | 0.18 | 36.2 | 7.6 | 53.8 |
| 8 [a] | 3.00 | 0.32 | 34.7 | 9.2 | 72.4 |

[a] star points; [b] cube points; [c] central points. $X_1$: extraction time. $X_2$: solvent-solid ratio.

**Table 2.** Quadratic equations fitting the experimental values of the 3 compounds extracted by UAE from cherries pulp.

| Compound | Equation | $R^2$ [a] | $R^2_{adj}$ [b] | Lack of Fit ($p$ [c]) |
|---|---|---|---|---|
| Cyanidin-3-O-rutinoside | $-327.29$ [d] $+ 2885.25$ [e] $X_2 - 5220.75X_2^2$ | 0.9249 | 0.8624 | 0.3002 |
| Quercetin-3-O-rutinoside | $-35.43 + 315.93X_2 - 505.70X_2^2$ | 0.7713 | 0.5806 | 0.4497 |
| *trans*-3-O-coumaroylquinic acid | $-291.93 + 32.23X_1 - 3.45X_1^2 + 2486.85X_2 - 4262.75X_2^2$ | 0.9354 | 0.8815 | 0.1616 |

[a] Coefficients of determination; [b] adjusted coefficients of determination; [c] significativity ($p < 0.05$); [d] intercepts; [e] regression coefficients. $X_1$: extraction time. $X_2$: solvent-solid ratio.

These results indicated that the models were more reliable in the case of cyanidin-3-O-rutinoside and *trans*-3-O-coumaroylquinic acid than quercetin-3-O-rutinoside one, which was expected considering that generally, the best hydroalcoholic mix for the extraction of flavonols requires a higher percentage of ethanol (i.e., 80%) [22].

$X_2$ was the main factor influencing the extraction of the three compounds, as confirmed by the significance ($p < 0.05$) of both linear and quadratic terms as well as their high regression coefficients (Table 2); as shown by the response surfaces which were generated based on the polynomial equations (Figure 3), a decrease of solid–solvent ratio by changing the volume corresponded to an increase of polyphenols yield probably due to the best swelling of the cherries powder which favored the cavitation process with the cell-wall disruption and more efficient mass transfer during the UAE [23]. However, this trend was observed till a value between 0.24 and 0.30 g/mL after which a drop in the concentrations of the compounds was registered; this optimal value was up to three times lower than in other research based on conventional extractions of sweet cherries [18,20] and up to eight times lower than that reported by Dumitraşcu et al. [24] for UAE of Cornelian cherry. It is worth noting that this finding (i.e., for our laboratory conditions 0.25 g sample in only 1 mL solvent mixture) could be particularly appreciable in terms of solvent saving and sustainability since recent trends in extraction techniques have largely focused on finding solutions that minimize the use of solvents by enabling process intensification [12]. Indeed, determining the optimal solid to solvent ratio is important in the development of extraction protocols not only to ensure efficient extraction of the desired compounds but also to reduce the waste of extraction solvent, which is of particular importance in large-scale extractions.

With regards to the $X_1$ factor, its linear and quadratic terms were significant just in the case of *trans*-3-O-coumaroylquinic acid, whose amount initially increased upon the raise of extraction time and reached a maximum level (at around 3 min), after which it started to decrease (Figure 3). Again, the fact that the phenolic compounds rapidly transfer to the

extraction solvent makes our method more economical than the methods developed for sour cherry phenols (100 min) [25] and sweet cherry anthocyanins (90 min) [18].

Finally, no significant interaction terms between $X_1$ and $X_2$ were evidenced, confirming how finding crossover effects between factors are not always common in the literature [14,18,26].

At this point, to search for the levels of the independent variables ($X_1$ and $X_2$) within the specified experimental range that produce the most desirable responses on the dependent variables (cyanidin-3-O-rutinoside, quercetin-3-O-rutinoside, and *trans*-3-O-coumaroylquinic acid), a desirability function was constructed by assigning to the predicted values a score ranging from 0 (very undesirable) to 1 (very desirable) through a general function optimization procedure. The graphs of desirability profiles are depicted in Figure 4.

Inspecting the profiles, the most desirable predicted responses (with a score of 0.83) on the dependent variables were simultaneously obtained at $X_1$ and $X_2$ levels of 3 min and 0.25 g/mL, respectively (Figure 3). It is worth noting that the optimized parameters allow the best simultaneous recovery of phenolic acids and flavonoids whose nutritional benefits, associated with their consumption either in the pulp or its extract, can potentially reduce the risk of health problems such as diabetes, cancer, cardiovascular diseases, inflammation, and Alzheimer's disease [1,27].

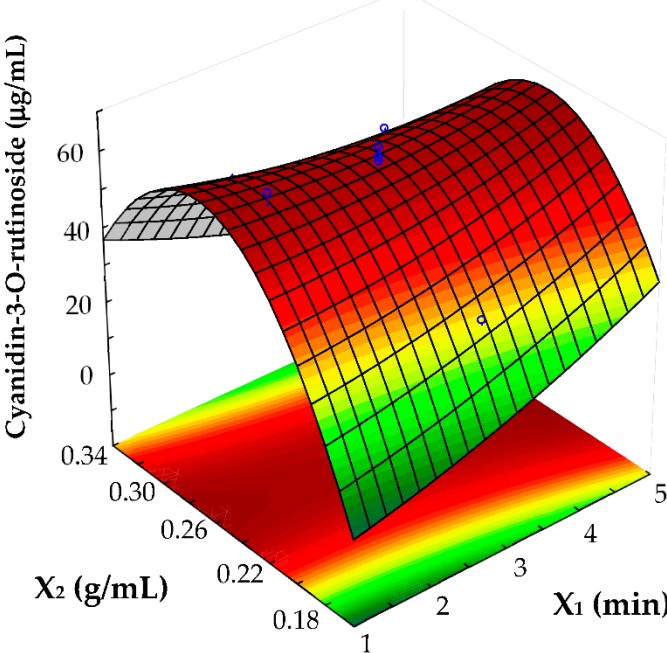

**Figure 3.** *Cont.*

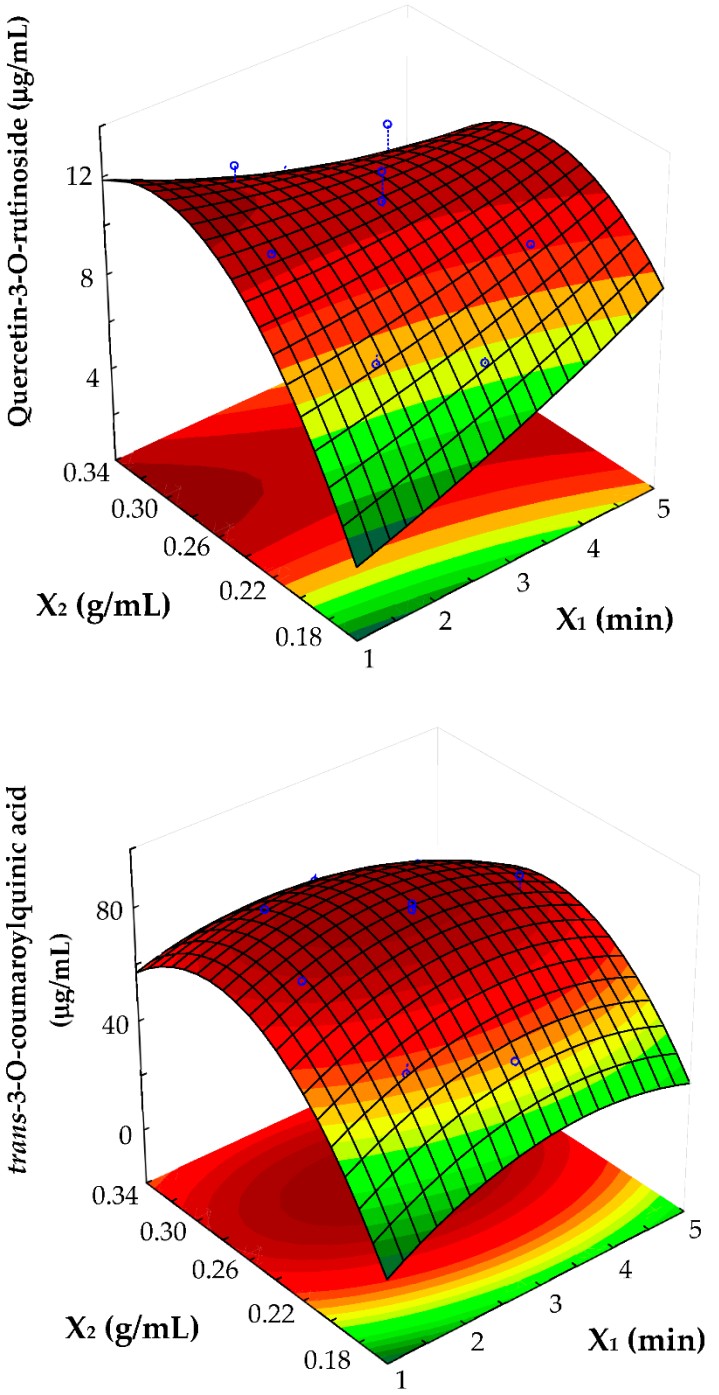

**Figure 3.** Response surface plots showing the effects of extraction time ($X_1$) vs. solid-solvent ratio ($X_2$) on polyphenols recovery from cherries pulp by UAE. Blue points retrace the CCD features.

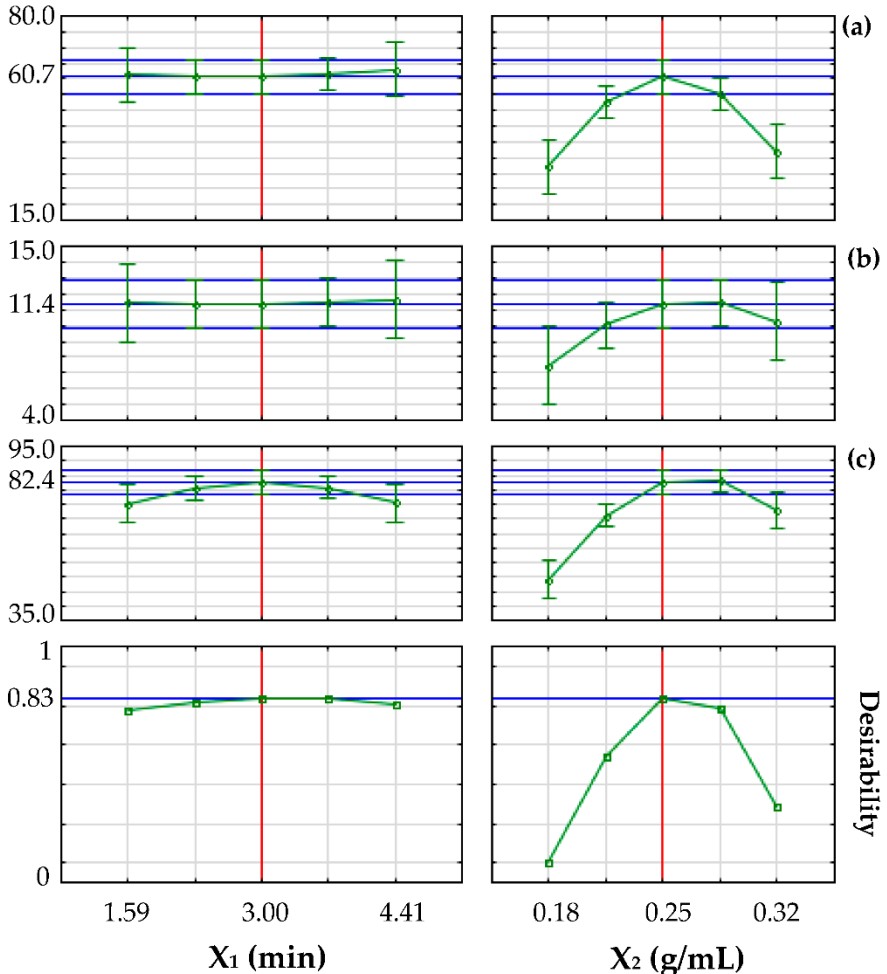

**Figure 4.** Desirability profiles of (**a**) cyanidin-3-O-rutinoside, (**b**) quercetin-3-O-rutinoside, and (**c**) *trans*-3-O-coumaroylquinic acid concentrations (µg/mL) in the extracts for each independent variable ($X_1$ and $X_2$). Blue and green lines indicate predicted and experimental value, respectively, while red lines indicate the best extraction conditions.

### 3.3. Validation Experiments

Definitely, the response optimization of UAE parameters for the extraction of polyphenols from sweet cherry (cv. Ferrovia) led to the following experimental conditions: $X_1$ = 3 min and $X_2$ = 0.25 g/mL in water/ethanol (1:1, *v/v*) and at set temperature (25 °C), sonication power (100 W), and sonication frequency (37 kHz). Therefore, to validate the predictive capacity of the gathered mathematical models, further experimental extractions were conducted in triplicates at the optimal conditions. As shown in Table 3, the quantities of cyanidin-3-O-rutinoside, quercetin-3-O-rutinoside, and *trans*-3-O-coumaroylquinic acid, estimated by further HPLC-DAD analyses, were in good agreement with the predicted values ($p > 0.05$, as determined by a *t*-test for single means), confirming the effectiveness and reliability of the RSM to establish the best UAE processing parameters. Even though the difference between the flavonol amounts (8.5 vs. 11.4 µg/mL) is almost significant ($p = 0.079$), as expected by the least efficient fitting shown in Table 2.

Finally, it is worth pointing out that the concentrations of cyanidin-3-O-rutinoside (240 mg/kg dw) and quercetin-3-O-rutinoside (320 mg/kg dw) expressed as mg/kg dry weight (dw) were higher and lower than those revealed by Iglesias-Carres et al. [20] in water/ethanol 80% extracts (29 mg/kg and 2200 mg/kg dw, respectively) of another sweet cherry variety. This finding was a major argument in favor of the aforementioned differential solubility of anthocyanins and flavonols in hydroalcoholic solvents.

**Table 3.** Validation test of the three polyphenols extracted under optimized UAE conditions.

| Compound | Experimental | Predicted | *p* |
|---|---|---|---|
| Cyanidin-3-O-rutinoside | 51 ± 9 | 60.7 | 0.228 |
| Quercetin-3-O-rutinoside | 8.5 ± 1.5 | 11.4 | 0.079 |
| *trans*-3-O-coumaroylquinic acid | 72 ± 10 | 82.4 | 0.213 |

## 4. Conclusions

This work focused on the green extraction procedure of polyphenols from sweet cherries pulp based on UAE, optimized through CCD and RSM tools, which provided good sustainability by excluding toxic solvents and reducing the extraction time. This approach could indirectly allow the development of scalable strategies (possibly extendable to other fruit matrices) from the laboratory to industry with a change of perspective centered on maximum effectiveness and efficiency but ensuring biosecurity requirements.

Additionally, the gathered findings would show how the followed procedure can support the increasing demand for natural products and extracts for food, cosmetic, and pharmaceutic sectors but reduce the overexploitation of plant resources.

**Supplementary Materials:** The following supporting information can be downloaded at: https://www.mdpi.com/article/10.3390/pr10081657/s1, Table S1. Chromatographic characteristics of identified compounds.

**Author Contributions:** Funding acquisition, project administration, writing—original draft: M.L.C.; project administration, data curation, writing—original draft, methodology, formal analysis: P.C.; writing—review & editing, supervision: F.C. All authors have read and agreed to the published version of the manuscript.

**Funding:** This study was supported by grant from the Apulian Region (Research for Innovation REFIN—POR Puglia FESR-FSE 2014/2020).

**Institutional Review Board Statement:** Not applicable.

**Informed Consent Statement:** Not applicable.

**Conflicts of Interest:** The authors declare no conflict of interest.

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
