# Peer review of "Optimization of a Green Extraction of Polyphenols from Sweet Cherry (Prunus avium L.) Pulp"

_processes, doi:10.3390/pr10081657_

Round 1
Reviewer 1 Report
Comments of the manuscript entitled “Optimization of a green extraction of polyphenols from sweet cherry (Prunus avium L.) pulp”.
The manuscript contains data on the green extraction of polyphenols from sweet cherry pulp. I suggest the following modifications.
Abstract needs rewriting, the lines from 11 to 16 describe general data, which are not necessary in the abstract. Delete. Start with ‘this work focused” (line 16).” The remaining abstract is satisfactorily presented.
Arrange the key words in the alphabetical sequence.
Introduction is unnecessarily lengthened and not primarily focused on the main thrust of the work. General information on sweet cherries, polyphenols, green extraction, organic solvents, energy consumption, solvent–solid ratio etc. have been amply and substantially reported and it serves no purpose to write again specifically in this manuscript. Precise writing in brief is advised Authors have stated their objective of the work clearly at the end of the introduction, which is sufficient, and needs no further elaboration.
Give the accession number and documented details of the experimental plant material.
What do you mean by “commercial maturity” (line 98)??Explain!
Is it necessary the section ‘2.2 Chemicals’ to add in the text??
Many of the experimental analytical methods followed here, are well standardized and widely known. It would be prudent to specify the protocols adopted which were deviated from the routine processes. This way prolonged text can be reduced.
Repetition of data in the main text and corresponding figures must be scrupulously avoided.
Figure 1. depicts CCD pattern, is confusing, what exactly is the significance??
Table 1 needs foot note to describe the table contents. Why experimental numbers were randomly written? How the time range of 2 to 4 minute was selected.? The values of three polyphenols are strikingly dissimilar, this requires further elucidation.
Table 2 also needs self-explanatory foot notes which are absent. Authors stated about the reduced extraction cost, but it is necessary to add the range of difference in the cost and how it was calculated.
How the values are predicted in the table 3, just by determining a t-test for single means? Give more information.
Figure 3. Desirability profiles, what do you mean by the term, “desirability profiles?”
The data on the response surface plots not undoubtedly interpreted, the accurate differences in the polyphenols have not been properly projected.
Table S1. RT = retention time; max = UV/Vis absorption maxima, what it advocates, add related description.
Conclusion needs shortening. The text in first two paragraphs is already stated in the earlier text.
The extraction procedures for food and nonfood products are not the same, this has to be considered. Bio safety is of utmost importance for selecting a solvent. Considering the nutritional quotients, what could be the nutritional benefits from a fruit pulp and from its extraction. It is recommended to add the related data and the references.
Overall, authors have extensive and convincing experimental data and presented satisfactorily and clearly. Their significant finding was in favor of the aforementioned differential solubility of anthocyanins and flavonols in hydroalcoholic solvents.
Author Response
Comments of the manuscript entitled “Optimization of a green extraction of polyphenols from sweet cherry (Prunus avium L.) pulp”. The manuscript contains data on the green extraction of polyphenols from sweet cherry pulp. I suggest the following modifications. Abstract needs rewriting, the lines from 11 to 16 describe general data, which are not necessary in the abstract. Delete. Start with ‘this work focused” (line 16).” The remaining abstract is satisfactorily presented. Arrange the key words in the alphabetical sequence.
According to your suggestions, the abstract was edited by deleting the general data, and keywords were rearranged in alphabetical sequence.
Introduction is unnecessarily lengthened and not primarily focused on the main thrust of the work. General information on sweet cherries, polyphenols, green extraction, organic solvents, energy consumption, solvent–solid ratio etc. have been amply and substantially reported and it serves no purpose to write again specifically in this manuscript. Precise writing in brief is advised. Authors have stated their objective of the work clearly at the end of the introduction, which is sufficient, and needs no further elaboration.
We have tried to significantly reduce the text while keeping the information that we consider important to describe the current state of the research field.
Give the accession number and documented details of the experimental plant material.
Analyzed sweet cherries did not come from an experimental field but from a commercial orchard located in Turi (longitude 40.56° E, latitude 17.12° N, and altitude 280 m). We have added the main characteristics of the cultivar in the revised manuscript (lines 101-105)
What do you mean by “commercial maturity” (line 98)??Explain!
Generally, the commercial maturity indicates that the fruit has reached the best organoleptic quality parameters (such as sweetness and acidity) which have a strong influence on consumer acceptance. Anyway, to avoid confusion we have rephased it as such: “Samples were harvested (second decade of June) once reached the best organoleptic quality parameters” (see lines 106-107 in the revised version).
Is it necessary the section ‘2.2 Chemicals’ to add in the text??
Usually, providing information about suppliers of the used chemicals is advisable. Reporting the supplier each time the chemical is cited in the text would create confusion and redundancy; therefore, we would prefer to maintain this section.
Many of the experimental analytical methods followed here, are well standardized and widely known. It would be prudent to specify the protocols adopted which were deviated from the routine processes. This way prolonged text can be reduced.
As requested, we have reduced the analytical methods when standardized or already known (see lines 108 -110 and lines 146 – 150 of the revised version).
Repetition of data in the main text and corresponding figures must be scrupulously avoided.
Accordingly, the repetition of the data regarding the desirable responses (already reported in Figure 3) has been deleted (see lines 472-473 of the revised version).
Figure 1. depicts CCD pattern, is confusing, what exactly is the significance??
The scheme reported in Figure 1 elucidates the characteristics of CCD in the chosen bidimensional variability field. We have changed “pattern” with “features” in the revised version.
Table 1 needs foot note to describe the table contents. Why experimental numbers were randomly written? How the time range of 2 to 4 minute was selected.? The values of three polyphenols are strikingly dissimilar, this requires further elucidation.
The experiment number reported in Table 1 refers to the trial conditions set by DoE, which are depicted in Figure 1. In order to avoid misunderstanding, we have added relative footnotes following your suggestion. Experimental numbers were randomly written to illustrate what we stated in the text (lines 165-167 and 280-281). As discussed in lines 271-273, we selected 2-4 min and 0.2-0.3 g/mL to allow a reduced energy and solvent consumption on the basis of previous screening experiments. Regarding the values of polyphenols, you are right their variation in that reduced experimental range impressed us, too. Actually, this was a further reason because we selected this restricted range.
Table 2 also needs self-explanatory foot notes which are absent. Authors stated about the reduced extraction cost, but it is necessary to add the range of difference in the cost and how it was calculated.
We have also added relevant footnotes to Table 2. Calculating the extraction cost is beyond the aim of this manuscript, therefore we corrected the statement as “to reduce the waste of extraction solvent” (see line 323 of the revised manuscript).
How the values are predicted in the table 3, just by determining a t-test for single means? Give more information.
The prediction equations (listed in Table 3), fitting the observed responses on the respective dependent variable, were used to obtain the predicted values for the dependent variables at any combination of levels of the predictor variables. While t-test for single means was used to compare the values of dependent variables from further experimental trials at the optimal extraction conditions with the predicted values for validation purposes.
Figure 3. Desirability profiles, what do you mean by the term, “desirability profiles?”
Response/desirability profiling allows inspecting of the response surface produced by fitting the observed responses using an equation based on levels of the independent variables. Using the Profiler tool (generating a report similar to that depicted in Figure 3) one can inspect the predicted values for the dependent variables at different combinations of levels of the independent variables, specify desirability functions for the dependent variables, and search for the levels of the independent variables that produce the most desirable responses on the dependent variables. The Profiler can be used to produce not only a prediction profile for a single dependent variable but also a compound prediction profile graph that shows the prediction profiles for multiple dependent variables. This can allow one to see whether the levels of the independent variables that maximize responses for one dependent variable also maximize responses on other dependent variables.
The data on the response surface plots not undoubtedly interpreted, the accurate differences in the polyphenols have not been properly projected.
Thanks to your observation, we noted the mistake of the data projection in particular referring to factor X1. Actually, we reported in the text a single value (0.25 g/mL), corresponding to the one obtained by the desirability function, instead of the range of the values at which the polyphenols recovery was maximum. Thus, we have corrected it as: “However, this trend was observed till a value between 0.24 and 0.30 g/mL” (line 313 of the revised manuscript).
Table S1. RT = retention time; max = UV/Vis absorption maxima, what it advocates, add related description.
We do not clearly understand your concern. However, retention time deals with the elution order of polyphenols (based on their dimension and polarity) in the chromatographic separation, which are detected through their UV/vis absorption at maxima wavelengths. These properties do not need to be specified because there are plenty of reports in the literature describing them.
Conclusion needs shortening. The text in first two paragraphs is already stated in the earlier text.
In the revised manuscript, we have tried to substantially shorten the conclusion section.
The extraction procedures for food and nonfood products are not the same, this has to be considered. Bio safety is of utmost importance for selecting a solvent. Considering the nutritional quotients, what could be the nutritional benefits from a fruit pulp and from its extraction. It is recommended to add the related data and the references.
This work focused on green extraction procedure since, in a perspective of sustainability, contemporary analytical chemistry aims at the development of analytical techniques that maintain, or even improve, traditional performance by excluding solvents considered toxic and/or disturbing for the simultaneous protection of operators and of the environment. This approach indirectly allows the development of scalable strategies from the laboratory to industry with a change of perspective that is based on the same principle: maximum effectiveness and efficiency ensuring all biosecurity requirements (see lines 505-510 of the revised manuscript). Regarding the nutritional effect, we have added the following text: “It is worth noting that the optimized parameters allow the best simultaneous recovery of phenolic acids and flavonoids whose nutritional benefits, associated with their consumption either in the pulp or its extract, can potentially reduce the risk of health problems such as diabetes, cancer, cardiovascular diseases, inflammation, and Alzheimer's disease.” (see lines 474-478 of the revised manuscript).
Overall, authors have extensive and convincing experimental data and presented satisfactorily and clearly. Their significant finding was in favor of the aforementioned differential solubility of anthocyanins and flavonols in hydroalcoholic solvents.
Thank you very much for your evaluation.

Reviewer 2 Report
I would like to thank the authors for their efforts in developing the extraction, quantification and characterisation of compounds extracted from the fruits of Prunus avium L. cv. Ferrovia.
I invite the authors to carefully review the entire manuscript to correct errors. In the following paragraphs, I will try to provide clear comments to improve the manuscript.
Title and abstract
The title is correct and the abstract is well ordered. I don´t have any consideration in this part.
1. Introduction
In my opinion, I think the introduction is correct, but I have been able to detect certain errors that I will comment below so that the authors can correct or improve them:
L47: I would suggest inserting an additional bibliographical note on extraction techniques: 10.3390/foods8070245.
L58: To better specify the reported ratio, is unclear. Make it scientifically friendly. It is not clear whether it is a ratio or a range.
L71: Temperature, in the case of UAE extractions involving all bioactive compounds from natural matrices, is a key parameter, which in the case of extractions of thermolabile molecules can play havoc in obtaining them. this is not highlighted in the introduction. I suggest exploring this concept in more detail.
L76-77: "approach; the latter." verify.
In the rest of the section, I have no corrections whatsoever.
2. Materials and methods
In general, the materials and methods are well structured and comprehensive. I will propose some modifications or corrections to the document:
L121: This proposed instrumentation does not appear to be the most suitable for performing UAE extractions while maintaining a constant temperature during the extraction time. As is well known, sonic waves produce a rise in the temperature of the medium, which could degrade the extracted thermolabile compounds. Argument.
L123: For how long? Please specify.
L145: "200 - 10 g/mL" I would reverse the two numbers for better understanding.
In the rest of the section, I have no corrections whatsoever.
3. Results and discussion
In general, the results are well expressed. However, I would suggest some changes or corrections in the different sections to improve the paper. I recommend that the authors provide a brief introduction in this section before starting to write directly about the different results.
L258: How is the temperature kept constant? Argument.
L276: "p" correct formatting in italics. If necessary, standardise the entire document (i.e. L291, L469, L472).
4. Conclusions
The conclusions are well structured. Considering the manuscript as a whole, I would try to elaborate on how the approach used in this study could be developed for other matrices or extraction techniques in the future.
References
References are clear and well structured.
Final Remarks
For improvement, the manuscript should be revised according to the above suggestions and those of other reviewers. In my honest opinion, I suggest a minor revision of the article. The authors have done work that provides interesting results.
Author Response
I would like to thank the authors for their efforts in developing the extraction, quantification and characterisation of compounds extracted from the fruits of Prunus avium L. cv. Ferrovia. I invite the authors to carefully review the entire manuscript to correct errors. In the following paragraphs, I will try to provide clear comments to improve the manuscript.
Thank you very much for your endorsement and useful suggestions.
Title and abstract: The title is correct and the abstract is well ordered. I don´t have any consideration in this part.
Thank you very much for your assessment.
- Introduction
In my opinion, I think the introduction is correct, but I have been able to detect certain errors that I will comment below so that the authors can correct or improve them:
L47: I would suggest inserting an additional bibliographical note on extraction techniques: 10.3390/foods8070245.
Accordingly, we have added this reference.
L58: To better specify the reported ratio, is unclear. Make it scientifically friendly. It is not clear whether it is a ratio or a range.
You are right the ratio reported in the original manuscript was unclear. Anyway, following the suggestion of reviewer 1, we decided to rephrase the entire part (see lines 52-55 of the revised manuscript).
L71: Temperature, in the case of UAE extractions involving all bioactive compounds from natural matrices, is a key parameter, which in the case of extractions of thermolabile molecules can play havoc in obtaining them. this is not highlighted in the introduction. I suggest exploring this concept in more detail.
In order to respect the request of reviewer 1 of shortening the introduction, we have chosen to add your useful suggestion in R&D section (lines 266-268, revised manuscript).
L76-77: "approach; the latter." verify.
We are sorry for the typing mistake. Of course, it has been deleted in the revised version.
In the rest of the section, I have no corrections whatsoever.
- Materials and methods
In general, the materials and methods are well structured and comprehensive. I will propose some modifications or corrections to the document:
L121: This proposed instrumentation does not appear to be the most suitable for performing UAE extractions while maintaining a constant temperature during the extraction time. As is well known, sonic waves produce a rise in the temperature of the medium, which could degrade the extracted thermolabile compounds. Argument.
Your observation is very relevant. Generally, it was difficult to maintain a constant temperature in UAE because of the reason you remembered. Fortunately, the short extraction time range (2-4 min) explored in this study minimizes this issue and allows to control the temperature around 25 °C (with a tolerance of ± 2 °), by adding a bit of crushed ice.
L123: For how long? Please specify.
Really, the extraction time was not specified because an object of optimization, as illustrated in Table 1.
L145: "200 - 10 g/mL" I would reverse the two numbers for better understanding.
Thank you for your observation. We have corrected the range (line 155).
In the rest of the section, I have no corrections whatsoever.
- Results and discussion
In general, the results are well expressed. However, I would suggest some changes or corrections in the different sections to improve the paper. I recommend that the authors provide a brief introduction in this section before starting to write directly about the different results.
L258: How is the temperature kept constant? Argument.
Fortunately, the short extraction time range (2-4 min) explored in this study minimizes this issue and allows to control the temperature around 25 °C (with a tolerance of ± 2 °), by adding a bit of crushed ice.
L276: "p" correct formatting in italics. If necessary, standardise the entire document (i.e. L291, L469, L472).
Correction done.
- Conclusions
The conclusions are well structured. Considering the manuscript as a whole, I would try to elaborate on how the approach used in this study could be developed for other matrices or extraction techniques in the future.
We have tried to interpret your suggestion, rephrasing the first part of the conclusions (lines 505-510).
References are clear and well structured.
Final Remarks
For improvement, the manuscript should be revised according to the above suggestions and those of other reviewers. In my honest opinion, I suggest a minor revision of the article. The authors have done work that provides interesting results.
Thank you very much for your valuable comments.

Reviewer 3 Report
Observations regarding the paper: Optimization of a green extraction of polyphenols from sweet 2 cherry (Prunus avium L.) pulp” authors: Maria Lisa Clodoveo et al.,
1. Abstract, line 16, it is necessary to insert the meaning for US.
2. Abstract, line 19, it is indicated to write „chromatographic analyses (HPLC-DAD).
3. It is not necessary to discuss in the Abstract (lines 11-14) and Introduction (lines 37 and 42) about the valorisation of waste from industrial fruit processing. The experimental analyses and statistical interpretation of the data were performed on fresh fruit.
Author Response
Observations regarding the paper: Optimization of a green extraction of polyphenols from sweet 2 cherry (Prunus avium L.) pulp” authors: Maria Lisa Clodoveo et al.,
- Abstract, line 16, it is necessary to insert the meaning for US.
As requested, we have specified “ultrasound” in the revised version.
- Abstract, line 19, it is indicated to write „chromatographic analyses (HPLC-DAD).
Correction done.
- It is not necessary to discuss in the Abstract (lines 11-14) and Introduction (lines 37 and 42) about the valorisation of waste from industrial fruit processing. The experimental analyses and statistical interpretation of the data were performed on fresh fruit.
Your observation is very relevant. Even though the method could efficiently work with processing leftovers, we only extracted fresh fruit. Thus, we have accepted your suggestion and deleted those parts.

Round 2
Reviewer 1 Report
Authors have revised the manuscript as suggested by the reviewer.